# FusPlace: The Fine-Grained Chip Placement with Hybrid Action Spaces and Feature Fusion

## Abstract

Chip placement is an essential and time-consuming step in the physical design process. Deep reinforcement learning, as an emerging field, has gained prominent attention due to its ability to replace weeks of expert model design. However, the current AI-based methods in chip design are still facing momentous challenges. We propose a fusion-based reinforcement learning framework to address the limited representation problem of both graph networks and CNN networks. Furthermore, the structure of PDQN in the hybrid action space allows for precise coordinate placement, compared to other RL-based structures in placement. The experimental results can demonstrate the effectiveness of our model.

## 1 Introduction

Physical design is crucial in Electronic Design Automation (EDA) (Markov et al., 2012) due to the exponential growth in scale and the increasing complexity of constraints. Placement is one of the most time-consuming stages in physical design. A well-executed placement process not only saves time but also ensures a successful final design outcome. The principle of 2D placement involves mapping amount macros and a millions of standard cells from a netlist to a specific location on a chip canvas, while minimizing total wirelength subject to constraints such as density and timing closure. A netlist is a structured data file that describes the logical relationships between components (such as standard cells, macros, etc.) in a chip and their interconnections (referred to as nets).

Placement approaches can be classified into two main categories: traditional placement methods and learning-based methods. Traditional placement techniques, which include simulated annealing (Vashisht et al., 2020), partitioning (Liu et al., 2019), and analytical algorithms (Lu et al., 2015), are all utilized in the field. In particular, analytical algorithms are widely used due to their robustness and quick computation abilities, as demonstrated by applications such as RePlAce (Cheng et al., 2018), ePlace (Lu et al., 2014), and DREAMPlace (Lin et al., 2019). Although traditional methods have achieved considerable progress, they still face many open challenges. On one hand, traditional placement techniques can lead to overlaps among components, which require resolution in the post-placement phase. On the other hand, common methods incorporating analytical approaches necessitate the modeling of optimization problems as differentiable functions to perform computations.

Methods based on deep learning and reinforcement learning are attracting increasing attention because they can leverage prior experience and simplify processes. In chip design, numerous RL-based approaches, such as those employed by Graph Placement (Mirhoseini et al., 2021), MaskPlace (Lai et al., 2022), and other researchers have achieved promising results by treating the placement problem as a sequential decision-making problem. However, these methods also have some inherent challenges and limitations.

Firstly, the conventional RL representation in chip design brings inconvenience and significant redundancy. For example, Graph Placement models chip location and connectivity information as a graph network (Scarselli et al., 2008), defined as $G = (V,E)$ where V represents the cells and E represent its nets. However, graph neural network(GNN) typically handle static features, and this approach requires passing all modeling information in each placement process. In reality, each placement is primarily influenced by the blocks connected to only its nets, and the global density only provides a simple constraint. Therefore, modeling unnecessary relationships between other networks results in significant redundancy in representation and computations.

Moreover, the placement problem in MaskPlace and ChiPFormer (Lai et al., 2023), which divided chip canvas into $224 \times 224$ and $84 \times 84$ grid sizes respectively, is solved by using CNN visual representations for circuit modules at the pixel level. When using a reinforcement learning model for action selection, each grid is considered equivalent to the center point of placing a macro. However, when dealing with large-scale components placement problem in industrial-level, the precision may be compromised. In the case of chip placement, inaccurate placement centers in local areas can lead to component overlaps, resulting in a failed physical design process.

To address the mentioned issue, we proposes a hybrid action space RL framework called Fusplace. By incorporating the original information of the chip, Fusplace enables more accurate chip placement and avoids issues such as component overlapping. By refining the definition of the action space, Fusplace can exert finer control over the placement position, thereby enhancing the precision and accuracy of the placement.

The highlight of our paper are:

- We use a hybrid action structure (P-DQN) to addresses the limitations of the original model's coarse-grained approach.
- We propose the feature fusion representation addresses the lack of interpretability in neural networks by decomposing netlist information into local and global parts and incorporating prior knowledge.

## 2 PRELIMINARIES

**Chip Placement**

The objective of placement achieved by minimizing HPWL (Chen et al., 2006) and subject to constraint which included congestion and overlap in the given netlist. The objective function is represented as follows.

$$\min_{x,y} WL(x,y) \quad \text{s.t. } Overlap(x,y) = 0 \text{ and } Cong(x,y) \leq C$$

**Markov Decision Process**

A standard Markov Decision Process (MDP) $\{ref\}$ is model as a quintuple $\mathcal{M} = \{\mathcal{S}, \mathcal{A}, \mathcal{P}, \mathcal{R}, \gamma\}$ where $\mathcal{S}$ is a state set, $\mathcal{A}$ is a action set, the transition function $\mathcal{P} : \mathcal{S} \times \mathcal{A} \times \mathcal{S} \to \mathbb{R}$, the reward function $\mathcal{R} : \mathcal{S} \times \mathcal{A} \to \mathbb{R}$ and $\gamma \in [0, 1)$ means the discounted factor. The policy function $\pi(a|s)$ maps the state $s \in \mathcal{S}$ into a action $a \in \mathcal{A}$. The action-value function $Q^\pi$ of policy $\pi$ is defined by

$$Q^\pi(s, a) = \mathbb{E}_\pi(R_t | s_0 = s, a_0 = a)$$

The aim of an RL agent is to find a policy to maximize the expected total discount reward

$$J(\pi) = \mathbb{E}_\pi[\sum_{t=0}^{T} \gamma^t r_t]$$

**Parameteried Deep Q-Networks**

Parameterized Deep Q-Networks (P-DQN)$\{ref\}$ extends the model from Markov Decision Process (MDP) to Parameterized Action Markov Decision Process (PAMDP) (Masson et al., 2016), and then build the architecture of P-DQN (Casgrain et al., 2022) to handle the problem of hybrid action space. The action space transformed by

$$A = \{(a, x_k) | x_k \in \mathcal{X} \text{ for all } a \in A_d\}$$

where $A_d = \{a_1, a_2, \cdots, a_k\}$ represents a discrete action set, and for each $a_i, i \in k$, there exists a corresponding continous parameter space $X$. We use a tuple $(a, x_k)$ describes the action. Then the related action-value function $Q^\pi$ of policy $\pi$ is devised by

$$Q^\pi(s, a, x_k) = \mathbb{E}_\pi(R_t | s_0 = s, a_0 = a, x_0 = x_k)$$

## 3 METHOD

### 3.1 MODEL ARCHITECTURE OVERVIEWS

When placing a macro module, the RL environment detects the chip nets make sure that other macros module is located and extracts all relevant macro units as local information. This information is then modeled as a matrix and fed into a feature extraction network. Then the feature network performs feature fusion on the original variable-length vector matrix, mapping it to a fixed-length 32-dimensional vector. The global information includes bin centers,chip meta-data and the congestion of each bin are separately fed into three MLP networks for feature extraction. The output results of all networks are concatenated to form an embedding vector. The policy network (DDPG and DQN) receives the embedding vector and output the hybrid action $(a, x_k)$. The related MDP specified as follows:

- State: An n-dimensional matrix containing information about the block and pin related to the current placement, as well as the global information under the current state. The advantage of multi-dimensional information representation in the state is that it allows for more accurate feature extraction and avoids the issue of the state space becoming too large when using feature fusion in the feature network, thus circumventing the need for CNN or GCN-based action representation.

- Action: The action space consists of a tuple $(a, x_k)$, which is formed by the combination of the continuous action space and the discrete action space. Here, $a$ represents the number of partitions in the grid, and $x_k$ represents the offset coordinates. This setup not only reduces the size of the action space but also allows for more precise placement centers to be obtained.

- Reward: To address the issue of sparse solutions, the reward consists of both internal rewards and a global reward based on the placement and congestion.

### 3.2 FEATURE EXTRACTION AND REPRESENTATION

Although GNN and CNN are effective feature extraction techniques, when it comes to chip placement problems. However, using GNN results in a large state space and significant information redundancy. Treating the problem solely as a computer vision task with CNN leads to insufficient network partitioning accuracy, ultimately resulting in failure in the overall physical design process. Hence, this paper proposes a solution by integrating the relationship between blocks and the network through feature fusion representation to address the problem found in previous works. The state space is divided into two categories: (1) local information and (2) global information. The overall flow of feature extraction is illustrated in Figure 1.

#### 3.2.1 LOCAL INFORMATION

The placement position of a newly placed block primarily depends on the placement of all the modules corresponding to the nets in the block.

To effectively model and capture the important information related to the placement of modules, we extract the most important 7-dimensional features for each block. These features include the center coordinates, width, height, area, and pin coordinates, which are represented as $x_d^i = (o_x^i, o_y^i, o_w^i, o_h^i, o_{area}^i, o_{pinx}^i, o_{piny}^i)$ to compose a matrix $X_d$. Since the blocks to be placed in the canvas correspond to different numbers of nets, and each net corresponds to a different number of placed blocks, our network architecture must be able to handle variable-length data.

To address this, we perform information fusion on the dynamically changing length in the feature space. We map the matrix $X_d$ to a high-dimensional embedding space $Z_l$ and sum it up, output a fixed 32-dimensional vector. The overall feature embedding obtained from local information is as follows:

$$Z_l = \frac{1}{n} \sum_{i=1}^{n} \psi_1(x_d^i) \tag{1}$$

### 3.2.2 GLOBAL INFORMATION

Global information consists of bin centers (coordinates of x and y in the bin) $X_{g1}$, bin densities (number of actions in each bin) $X_{g2}$, and meta-data $X_{g3}$.

The global information is individually mapped into 32 dimensional vectors using MLP networks with network parameters $\psi_2, \psi_3$ and $\psi_4$. This approach helps to extract information more accurately.

### 3.3 DRL MODEL FOR THE PROCESS OF CHIP PLACEMENT

This section introduces the rationale and structure of using P-DQN to solve the chip placement problem. Traditional RL methods for chip placement divide the canvas into grids and treat the placement action as a discrete action space. The advantage of this approach is that the canvas can be directly considered as an image and fed into the RL policy network, benefiting from the use of computer vision techniques for problem solving. However, when dealing with real chip placement problems on a large canvas, simply treating the center as a single grid leads to low accuracy and fails to address issues like chip overlapping in later stages of placement.

Treating the canvas as a continuous action space, on the other hand, may cause the DDPG algorithm (Silver et al., 2014) to get stuck in a local optimum due to its deterministic policy and updates based on a local value function, making it difficult to converge. P-DQN overcomes this challenge by simultaneously producing discrete and continuous action values. The canvas is initially divided into n regions as the discrete action space, with each action representing the coordinate of the center of a region. The continuous action space is then considered as an offset for the centers of the discrete actions. The combination of the two parts represents the coordinates of placement macros.

The Parameterized DQN (P-DQN) combines the structures of DQN and DDPG. In the DDPG framework, the actor network with parameter $\theta_1$ maps the input state $X_d$ to a continuous parameter vector $x_k$, which represents the displacement of placing the corresponding discrete position coordinates. The DQN network, on the other hand, acts as a discrete action network, mapping the concatenated state and $x_k$ to the discrete action space. The resulting hybrid action is represented as $(k, x_k)$, where $k$ represents the chosen action in the discrete action space, and $x_k$ represents the corresponding offset for the chosen action.

The related Bellman euqation as

$$Q(s_t, a_{d_t}, a_{c_t}) = \mathbb{E}_{r_t, s_{t+1}}[r_t + \gamma \max_{a_{d_t} \in \mathcal{A}_\lceil} \sup_{a_{c_t} \in \mathcal{A}_\rfloor} Q(s_{t+1}, a_d, a_c)|s_t = s, A_d = a_d, A_c = a_c]$$

The Q-network and DDPG-network are trained on batch sample form a replay buffer R. The loss Q-network function of the defined as:

$$\mathcal{L}_t^Q(\theta_1) = \frac{1}{2}(Q_{\theta_1}(s_t, a_d, a_c) - y_t)^2$$

The DDPG-network The loss Q-network function of the defined as:

$$\mathcal{L}_t^d(\theta_2) = -\sum_{k=1}^{K} Q(s_t, a_d, a_c; \theta_1)$$

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
