# OpenReview forum: "The Fine-Grained Chip Placement with Hybrid Action Spaces and Feature Fusion"
_ICLR.cc/2024/Conference — Submitted to ICLR 2024_

### Official Review · Reviewer_XWnU · 2023-10-17

**Soundness:** 1 poor
**Presentation:** 1 poor
**Contribution:** 2 fair
**Rating:** 1
**Confidence:** 5

**Summary:**

The authors proposed a hybrid action space for the problem of chip placement using deep reinforcement learning. They also proposed using a simpler architecture compared to previous GNN- and CNN-based architecture for the agent’s model. The argued that using a hybrid discrete and continuous action space (denoting the bin and the offset from the center of the bin) help to avoid overlap and overall accuracy of the placement. However, the paper is not complete and it doesn’t have experimental results and the method is not presented completely.

**Strengths:**

- Presented a novel idea for hybrid action space for chip placement problem.

**Weaknesses:**

- The paper is incomplete and doesn’t have experimental results and the method is not presented completely.
- The paper includes grammatical errors.

**Questions:**

None

---

### Official Review · Reviewer_YRSP · 2023-10-20

**Soundness:** 1 poor
**Presentation:** 1 poor
**Contribution:** 1 poor
**Rating:** 1
**Confidence:** 5

**Summary:**

The paper wants to present fusion-based reinforcement learning for chip placement tasks.

However, the paper should be rejected because (1) it seems an unfinished work with only five pages. It does not contain any experimental results or any figures. (2) The proposed method lacks novelty, just like the extension of features and change of backbones of RL. (3) The presentation is poor.

**Strengths:**

Chip placement is a small field of applied research, it can further extend the application areas of AI.

**Weaknesses:**

It seems an unfinished work without experimental results and figures.

**Questions:**

1. Why the paper does not include experimental results?

2. It mentions Figure 1 in Sec. 3.2. But where is it?

---

### Official Review · Reviewer_WddF · 2023-10-27

**Soundness:** 2 fair
**Presentation:** 1 poor
**Contribution:** 2 fair
**Rating:** 1
**Confidence:** 4

**Summary:**

This paper proposes a fusion-based reinforcement learning framework for chip placement called FUSPLACE which extracts both local and global information. Specially, Parameterized DQN is adopted to produce discrete and continuous action values as precise placement coordinate.

**Strengths:**

1. The novelty lies in the design of hybrid action structure that divides the chip canvas into discrete bins combined with a continuous offset from the centers.

**Weaknesses:**

1. The expression of the paper is confusing, as most variables in the equation are not clearly explained and reference placeholders are still left.
2. The reviewer feels this paper is only half-baked, since there are no experimental results to validate the effectiveness of proposed structure.

**Questions:**

1. Why the continuous parameter vector is decided before discrete position coordinates?
2. Could the authors explain the details of internal rewards?
3. Have you conducted any ablation studies to evaluate the impact of different components of the proposed model?

---

### Official Review · Reviewer_QMCZ · 2023-11-01

**Soundness:** 1 poor
**Presentation:** 1 poor
**Contribution:** 1 poor
**Rating:** 1
**Confidence:** 3

**Summary:**

This seems to be a withdrawn submission.

**Strengths:**

N/A

**Weaknesses:**

N/A

**Questions:**

N/A

---

### Meta-Review · Area_Chair_3rge · 2023-12-06

**Metareview:**

The paper is incomplete and doesn’t have experimental results and the method is not presented completely. All reviewers agreed that the expression of the paper is confusing, as most variables in the equation are not clearly explained and reference placeholders are still left. Therefore, I opt for the rejection.

**Justification For Why Not Higher Score:**

Please see the meta-review

**Justification For Why Not Lower Score:**

N/A

---

### Decision · Program_Chairs · 2024-01-16

Reject